# *“So, if she wasn’t aware of it, then how would everybody else out there be aware of it?”*—Key Stakeholder Perspectives on the Initial Implementation of Self-Collection in Australia’s Cervical Screening Program: A Qualitative Study

**DOI:** 10.3390/ijerph192315776

**Published:** 2022-11-27

**Authors:** Claire M. Zammit, Nicola S. Creagh, Tracey McDermott, Megan A. Smith, Dorothy A. Machalek, Chloe J. Jennett, Khic-Houy Prang, Farhana Sultana, Claire E. Nightingale, Nicole M. Rankin, Margaret Kelaher, Julia M. L. Brotherton

**Affiliations:** 1Centre for Health Policy, Melbourne School of Population and Global Health, University of Melbourne, Melbourne, VIC 3010, Australia; 2Australian Centre for the Prevention of Cervical Cancer, Melbourne, VIC 3010, Australia; 3The Daffodil Centre, The University of Sydney, a joint venture with Cancer Council NSW, Sydney, NSW 2011, Australia; 4The Kirby Institute, Wallace Wurth Building, University of New South Wales Kensington, Sydney, NSW 2052, Australia; 5Centre for Women’s Infectious Diseases, The Royal Women’s Hospital, Melbourne, VIC 3052, Australia; 6National Cancer Screening Register, Telstra Health, Melbourne, VIC 3000, Australia

**Keywords:** self-collection, self-sampling, cervical screening, implementation science, qualitative, experience, women’s health, primary care, preventative health, health services, cancer prevention

## Abstract

Background: In December 2017, the Australian National Cervical Screening Program transitioned from 2-yearly cytology-based to 5-yearly human papillomavirus (HPV)-based cervical screening, including a vaginal self-collection option. Until July 2022, this option was restricted to under- or never-screened people aged 30 years and older who refused a speculum exam. We investigated the perspectives and experiences of stakeholders involved in, or affected by, the initial implementation of the restricted self-collection pathway. Methods: Semi-structured interviews were conducted with 49 stakeholders as part of the STakeholder Opinions of Renewal Implementation and Experiences Study. All interviews were audio recorded and transcribed. Data were thematically analysed and coded to the Conceptual Framework for Implementation Outcomes. Results: Stakeholders viewed the introduction of self-collection as an exciting opportunity to provide under-screened people with an alternative to a speculum examination. Adoption in clinical practice, however, was impacted by a lack of clear communication and promotion to providers, and the limited number of laboratories accredited to process self-collected samples. Primary care providers tasked with communicating and offering self-collection described confusion about the availability, participant eligibility, pathology processes, and clinical management processes for self-collection. Regulatory delay in developing an agreed protocol to approve laboratory processing of self-collected swabs, and consequently initially having one laboratory nationally accredited to process samples, led to missed opportunities and misinformation regarding the pathway’s availability. Conclusions: Whilst the introduction of self-collection was welcomed, clear communication from Government regarding setbacks in implementation and how to overcome these in practice were needed. As Australia moves to a policy of providing everyone eligible for screening the choice of self-collection, wider promotion to providers and eligible people, clarity around pathology processes and the scaling up of test availability, as well as timely education and communication of clinical management practice guidelines, are needed to ensure smoother program delivery in the future. Other countries implementing self-collection policies can learn from the implementation challenges faced by Australia.

## 1. Introduction 

Australia is a leader in the prevention of cervical cancer, being an early adopter of a National Cervical Screening Program (NCSP) in 1991 and a National Human Papillomavirus (HPV) Vaccination Program in 2007 [1,2] with the introduction of the NCSP alone halving cervical cancer cases and deaths [3]. Although since 2002, a steady plateau in the reduction of cases has been observed [4]. This impart is due a long-term decline in cervical screening participation [5] with just over half of the eligible population screening over a 2-year period as recommended [6]. Critically, some groups in the Australian population have lower screening participation rates and therefore a higher cervical cancer burden, including those people who live in rural and remote Australia or in socioeconomically disadvantaged areas, Aboriginal and Torres Strait Islander populations, and some migrant populations [6,7,8,9,10]. This is a cause of significant inequity, with the large majority (~75%) of cervical cancers diagnosed in people who are not up to date with screening [11]. Australia has a low overall cervical cancer incidence rate of 6.3 cases per 100,000 women [2]. This is nearing the World Health Organization’s (WHO) cervical cancer elimination goal (<4 cases per 100,000) [12]. Modelling data indicates that Australia is on track to eliminate cervical cancer by 2028–2035, assuming the maintenance of high current nonavalent HPV vaccination coverage (~80% completed course) and successful transition from cytology based to HPV-based screening [1]. However, masked inequities in screening participation may compromise Australia’s capacity to achieve and sustain the elimination goal for cervical cancer. More critically, there is an unacceptable risk of leaving behind the groups who bear the greater burden of this cancer [2].

In December 2017, Australia’s NCSP underwent a ‘renewal’ (rNCSP), changing from two-yearly Papanicolaou testing starting at age 18–20 years to five-yearly primary HPV-based testing starting at age 25 years. This transition was informed by international evidence about the superior performance of HPV-based screening in the prevention of cervical cancer [13]. These changes provided the opportunity to offer self-collection where a person uses a flocked swab to take their own HPV sample from the low-mid vaginal cavity [5]. Traditional cervical screening requires participants to undergo a speculum examination conducted by a practitioner to visualise the cervix to obtain a cervical sample for testing. Barriers to completing a clinician-collected cervical screening test are well documented including feeling a sense of embarrassment or pain [14] cultural reasons [15], fear of the procedure or results [16], a history of sexual trauma or abuse [17] or lack of time to prioritise preventative health [18]. Self-collection overcomes many of these barriers, is just as effective in detecting cervical precancer as clinician-collected tests, has been shown to increase participation and is highly acceptable to screening participants [19,20]. Previous Australian modelling found that if self-collection enabled even one lifetime screen, this would reduce a woman’s risk of cervical cancer by around 40% [3]. All these factors make self-collection an attractive option to engage those who do not currently participate in clinician-collected screening. 

Self-collection has been universally available in the renewed NCSP (rNCSP) since July 2022 [21,22] and is delivered via a practitioner-supported model of care, meaning self-collection needs to be ordered and overseen by a healthcare professional (mostly within the context a primary care consultation). This differs from models of care that have been used in some other international settings such as mail-out models in high-income countries, where self-collection kits are mailed to eligible participants, or models where testing is provided during home visits via health workers [19]. When initially introduced in Australia, self-collection was restricted to people aged 30 years or more, who were at least two years overdue for screening or those who had never-screened and had declined a clinician-collected screening test [21]. However, a commercially available test was not apparent during its initial introduction meaning regulations required laboratories to validate HPV testing on self-collection devices. This significantly delayed the full introduction of self-collection with only one laboratory in the country having validated self-collection testing on a flocked swab at that time [23].

This therefore study aimed to elicit the opinions and experiences of key stakeholders in relation to the introduction of the restricted self-collection cervical screening pathway within Australia’s NCSP, as part of a broader STakeholders Opinions of Renewal Implementation and Experiences Study (STORIES), which aimed to document stakeholders’ experiences, barriers and facilitators in implementing the renewed NCSP more broadly. 

## 2. Methods 

Study Design and Recruitment: STORIES utilised a qualitative approach to explore stakeholders’ perspectives regarding the implementation of the rNCSP, including self-collection. In brief, semi-structured interviews (Appendix A) were conducted with 49 stakeholders who were directly or indirectly involved in the implementation of the rNCSP in Australia. This included State and Territory program managers and other key stakeholders in program delivery including cervical screening providers, specialists, representatives from peak bodies, and laboratory service providers. A sampling frame of all potential stakeholders (n = 87) was collated by the authorship team given their extensive networks and relationships across the sector. Potential participants were included based on their role in the rNCSP, direct or indirect involvement in the implementation process, their knowledge or expertise and geographic location. Sampling was completed with a view to achieve to broad diversity in participant location and role in the rNCSP. Potential participants were sent via email an invitation letter, a plain language statement describing the research and their role in the study. 

Data Collection & Analysis: Written informed consent was obtained for participation in the study. Interviews were conducted between November 2018 and August 2019, eleven to twenty months into self-collection’s initial implementation. Interviews were audio recorded and transcribed. 49 participants represented over half the original sampling frame. The STORIES research team believed 49 interviews generated sufficient data to reflect consistency across the themes deduced and data saturation. No financial incentives were provided to participate. Interviews followed a guide developed by the authors using the Conceptual Framework for Implementation Outcomes described by Proctor, which specifies eight distinct implementation outcomes: *acceptability, adoption, appropriateness, feasibility, fidelity, implementation cost, penetration and sustainability*; which were operationalised as described in Table 1 [24] (Appendix A). Thematic analysis, using a combination of inductive and deductive coding, was utilised to analyse and interpret emergent themes in interviews using NVivo11 [25]. 

## 3. Results 

A total of 87 stakeholders were invited to participate in the study with 49 agreeing to be interview (56%). 36 stakeholders declined to participate or did not respond to invitation. Professional groupings for the 49 study participants is summarised in Table 2. It should be noted that 10 participants were counted in two or more categories because they had more than one professional grouping, and that advisory committee members typically also fit into another category. The main implementation outcomes described in this study are acceptability, appropriateness, feasibility, fidelity, adoption, and penetration. [24]. No themes related to implementation costs and sustainability were deduced as these outcomes are related to late-stage program implementation as the focus of our research was initial implementation. 

### 3.1. Acceptability and Appropriateness of Self-Collection

Stakeholders found the concept of self-collection acceptable and viewed it as an appropriate and exciting opportunity to engage or re-engage never- or under-screened populations. Primary care providers interviewed who had utilised the self-collection pathway, found that it was a useful strategy to encourage cervical screening participation among people who would not otherwise participate. 


*But now I find the opportunity to say to the women, “Well look, if you don’t want to do that, we can have a self-collected option.” It just gives you that extra opportunity to maybe engage that woman. I think that’s a positive for the program, just having that additional opportunity to capture an under-screened woman.*
(Participant 40, Healthcare Provider)

There was a strong sense of optimism and support from all stakeholders for the introduction of self-collection within the Australian rNCSP. The introduction of self-collection was regarded as a great opportunity to provide eligible participants with an alternative to a speculum examination. 


*I think [that] self-collection is fabulous, and I’m really looking forward to that being the policy, that all women can have that all the time. And I think that will just be a deal-breaker for cervical screening participation, and we’ll actually go to a point where we’ve got close to 100%.*
(Participant 37, Program)

Whilst self-collection was perceived as a potentially more acceptable option for reaching specific populations who are under-screened, stakeholders reflected a need for a strong focus on consistent cultural safety throughout the full screening and follow-up pathway for self-collection. For example, one participant noted when a triage test or colposcopy were required: 


*If a woman does a self-sample … and at the moment it’s pretty hard to do a self-collect anyway, but if a woman who is in [the Northern Territory] doing a self-collect and it comes back positive, how do they return to their provider for the cervical screening [LBC sample], and then colposcopy if that’s required? So we still, for these groups, really need to address the cultural safety and appropriateness, the access to a culturally appropriate service, so that they do participate in the program.*
(Participant 43, Policy/Advocacy)

### 3.2. The Feasibility and Fidelity of Self-Collection 

Participants expressed that the most significant barrier to the feasible implementation of self-collection and its continued use was that it was not offered by all pathology providers (and only one accredited to process self-collection samples in Australia at the time of initial implementation). Pathology labs were required to undertake their own validation and accreditation in testing self-samples as there was no commercially available approved self-collection test registered for use in Australia. With only one laboratory having completed this accreditation at initial implementation, it had been assumed that this would facilitate the test’s use by other laboratories. However, the regulator (Therapeutic Goods Administration in Australia [23]) advised that this was not the case. This was perceived as a major barrier for pathology labs to offer self-collection testing. Stakeholders reported lack of clarity surrounding this issue which led to misinformation that self-collection was only available in one Australian state. This in turn impacted practitioner uptake and resulted in missed opportunities to engage screen eligible participants. 


*But it’s very hard, I guess, for a Victorian laboratory to spread the message nationally when some of the other pathology services, when practitioners ask them, are just simply saying, “It’s not available,” which is not actually true.*
(Participant 7, Policy/Advisory)


*So what that meant was you had women presenting, having heard about self-sampling and their practitioners believing they couldn’t send us a sample, sending those women away. And that’s lost opportunity….*
(Participant 4, Pathology sector)

There was a deliberate absence of centralised program promotion of self-collection as it was not universally available, and especially once it became clear that only one laboratory would be able to offer it (at least in the short term, depending when/if other laboratories decided to apply for accreditation or act as conduits to testing). This was a major factor limiting promotion. 


*So we [were] quite on top of the not promoting it widely anyway, we actually quite consciously removed a lot of information to do with self-collection because it simply wasn’t going to be able to be provided. But of course, a lot of healthcare providers were aware that it was part of what the renewal program was going to be able to offer. So there was a lot of disappointment from the sector… This is seen as a really great missed opportunity for the program.*
(Participant 36, Program)

As a result, the awareness and use of self-collection was limited. Primary care providers displayed disappointment in the limited promotion and subsequent low adoption amongst the wider sector given the perceived potential of self-collection in engaging screen-eligible participants. 


*And I know in one instance where there was a GP who has been involved in a lot of this work, who was really concerned that she wasn’t aware of it. So, if she wasn’t aware of it, then how would everybody else out there be aware of it? So, I know it’s been a bit contentious about the information, about, out to clinicians.*
(Participant 6, Other)


*The self-collection has been interesting. I have been disappointed in how little that’s been promoted in my circles because I just think it’s such a wonderful alternative and we really should be, I mean … about you know, sort of 50 or at best 60% of women who are regularly screened for cervical cancer which means that we should really be offering self-collect to 40% or all women. And that’s just not happening.*
(Participant 34, Healthcare Provider)

### 3.3. The Impact of Promotion on Adoption and Penetration of Self-Collection

The adoption and penetration of self-collection into primary care was impacted by a lack of perceived promotion and unclear communication from a program/system level. Self-collection was perceived as being an important mechanism to increase screening participation amongst populations with a diverse range of reasons for not participating in screening. It was viewed as being able to address psychosocial barriers to screening, such as the experience of sexual violence or history of trauma. Stakeholders also perceived that self-collection was a way of respecting the cultural values of screen-eligible participants who identified as being Aboriginal and Torres Strait Islander or were from Culturally and Linguistically Diverse populations. 


*I think as I mentioned earlier that not having self-collection available has been an issue for encouraging participation in Aboriginal and culturally and linguistically diverse background women because we know that women from these population do tend to be over-represented in the under-screening data. So self-collection is certainly an avenue that we make.*
(Participant 36, Program)

Confusion regarding the availability of self-collection, particularly amongst different Australian states, impacted the adoption of self-collection into clinical practice as did variability in awareness of the pathway amongst primary care practitioners. This extended to confusion surrounding the following issues:Pathology processes of self-collection

There was a lack of information and misinformation around which pathology providers could receive samples to test. 


*Then we had to order the swabs and it’s like, “what swab are we ordering? Oh we haven’t got them yet because we don’t know which lab we’re gonna send it to”. So there was a delay there just ‘cause you know, knowing Victoria was accredited, knowing which swab to order, getting added to the order form, like the pathology order form, like it wasn’t on there.*
(Participant 8, Healthcare Provider)

Self-collection’s test accuracy despite updated evidence

Broader adoption of self-collection was impacted by practitioners’ confidence in utilising self-collection. There was no communication regarding updated evidence displaying equivalence in sensitivity between clinician-collected and self-collected tests. 


*Providers] didn’t understand the science behind it. So, that was … they didn’t feel confident. I think that was the thing with that, although I have heard more on one occasion that they [providers] definitely don’t believe the science of self-collection. Some clinicians, not all. Like, talking a couple, have said to me, “I would never recommend that, ‘cause I don’t trust that that test is as good as me doing it… I would never recommend it.”*
(Participant 13, Research)

Self-collection’s eligibility criteria and interpretation of guidelines

Practitioners’ uncertainty regarding eligibility of self-collection was common, from the pathology sector perspective and resulted in the rejection of samples. 


*And then there was a confusion of who is eligible for self-collect. That is one big thing. Not only the women themselves, but the clinicians, they think, “Oh, that’s a good idea, so we’ll self-collect.” All women they are all under 30, so we can’t do it, and we are bound by the NATA.*
(Participant 18, Pathology sector)

Additionally, difficulties accessing screening histories needed to assess eligibility within a suitable timeframe impacted implementation of self-collection and confidence in the National Cancer Screening Register. [26]


*I mean we’ve all had a few where the woman has declined a speculum examination and said, “Oh yeah, it’s definitely more than four years” and we’ve had the swab returned saying it’s two weeks early we can’t process it …. I’ve given up on calling the register to clarify that in real time, I don’t have time … so it would be nice when that’s electronic and available to providers eventually.*
(Participant 28, Healthcare Provider)

## 4. Discussion 

This qualitative study explored key stakeholder experiences and opinions regarding the initial introduction of self-collection for cervical screening as a part of the Australian rNCSP. Most stakeholders viewed the introduction of self-collection as a highly acceptable and appropriate modality to increasing screening participation given the sense of autonomy it can provide to people who experience psychosocial, cultural, or physical barriers to screening. However, the fidelity and feasibility of pathway implementation was significantly impacted by the lack of a commercially available test and accredited laboratories to test self-collected samples. Our findings also demonstrate that the adoption and penetration of self-collection within the health system was significantly impacted by limited communication or concerted efforts to promote self-collection, resulting in missed opportunities and misinformation regarding the pathway’s availability. 

Overall, stakeholders were positive regarding the introduction of self-collection within the program and viewed it as an exciting opportunity to improve screening participation, consistent with healthcare provider attitudes reported in other studies [27,28]. However, our research adds further knowledge by documenting specific implementation challenges relevant to the self-collection at both the health service and system level. The absence of health system direction and clarity surrounding the nuances of the introduction of the self-collection policy ultimately led to misconceptions surrounding its availability. Multiple stakeholders interviewed considered the perceived lack of availability of self-collection as the biggest barrier to implementation and ongoing utilisation. From providers’ perspectives, this significantly impacted their ability to offer self-collection opportunistically and led providers to believe that self-collection was only available in one Australian State [29]. This was further compounded by the fact that self-collection was not available until January 2018. Ensuring all sectors that are tasked with implementing self-collection are adequately prepared and informed regarding the nuances of the introduction of self-collection to a NCSP is crucial to ensure continuing provider acceptability and utilisation. Collectively, the initial preparation and introduction of self-collection as a new evidence-based innovation to improve participation; was perceived as having a lack of health system guidance and transparency. This had significant impacts on the uptake self-collection particularly within primary care providers who are the key implementers of self-collection. As demonstrated by our study, feasibility and fidelity issues mainly resulted from an unexpected regulatory delays as self-collection was not listed as for “intended use” by HPV test manufacturers, which under Australian regulations required laboratories to conduct their own in-house validation. This meant that providers who had spent time engaging screen-eligible participants to self-collect, were told samples could only be tested if sent to the one laboratory on their approved testing device [23]. By June 2022, three laboratories within Australia had gained accreditation to test self-collected samples as part of the NCSP [30]. Two manufacturers now have an intended use indication registered for its HPV test in Australia with others expected to follow shortly. This will ensure that most laboratories will be able to process these specimens as Australia introduces universal access to self-collection, with the expectation that those who cannot forward samples to those who can. The impact of a lack of a commercially available test on the feasibility and fidelity of the implementation of self-collection warrants further investigation in the Australian context, and will likely provide learnings for other health systems wanting to implement universal self-collection as a primary screening method.

Our study underscores the importance of having a fully functioning registry to support program implementation prior to introduction. In the Australia context, this was of particular relevance for the self-collection pathway at the time of its initial implementation when it was restricted to under-screened people. Practitioners faced challenges in accessing timely and accurate cervical screening histories to assess self-collection eligibility, which contributed to a general confusion around eligible and potential missed opportunities for screening. The National Cancer Screening Register now has a healthcare provider portal with integration with some commonly used practice management software, which may overcome this sort of issue in future (and it is now less of a barrier as there is universal access to self-collection). It is important that the system level issues reported by our study and elsewhere, which may have contributed to a general confusion and mistrust of self-collection [27,31,32] are successfully addressed and practitioner attitudes changed moving forward. 

We demonstrated that the lack of promotion of self-collection not only contributed to low adoption and support for implementation amongst primary care, but also to the misconception by some that self-collection is an inferior test in comparison to clinician-collected [27,28,29,31]. Very limited promotion regarding the availability of self-collection to both consumers and health care providers has very likely contributed to a substantially low adoption of restricted self-collection in the first two years of the renewed program [33]. Among primary care practitioners, there appears to be the strong misconception that self-collection is inferior compared to a clinician-collected HPV samples [27,31,34] with similar findings have been documented elsewhere [28,29,32]. At the time of the policy decision in Australia (in around 2014), the early evidence suggested that, while HPV testing on a self-collected sample had a greater sensitivity than Pap testing, it was slightly inferior to HPV testing on a clinician-collected sample, because many of the studies had used HPV tests based on signal-amplification assays rather than PCR-based assays [35]. Therefore, to avoid compromising the overall effectiveness of the program, self-collection was restricted to under-screened people [3]. Since then, an updated systematic review has shown that PCR-based HPV tests perform equally well on self-collected and clinician-collected samples (as a result, only PCR-based tests can be used on self-collected samples in Australia) [19,36]. Additionally, modelling demonstrated that, even a slightly less sensitive test would still result in a net benefit to the program if it increased participation [37]. This evidence has informed the policy change to make self-collection universally available [22]. If this misconception continues, it has the potential to further reduce provider and consumer trust in the program. It is therefore imperative that there is frequent and persistent messaging from a program policy level promoting self-collection as an available tool that is as accurate as clinician-collected screening,

The strengths of our analysis include the diverse range of stakeholders including primary care practitioners, pathology providers and health system/program stakeholders who shared their experiences about self-collection, rather than clinicians only, and the use of an established implementation framework to comprehensively consider the data. We purposively sampled stakeholders based on their role in implementing self-collection as a part of the rNCSP and location to represent a diverse range of opinions and experiences. Limitations of this study include that the issues documented only reflect early challenges (first 17 months of initial implementation) experienced by providers, policy and program stakeholders regarding self-collection cervical screening. The rNCSP is now within its 5th year and there is no evidence to suggest these issues have been resolved or persisted. The implementation of self-collection is ongoing and experiences, and issues are likely to evolve over time, especially given the transition to universal access to self-collection. We also did not capture the experiences of screen eligible people who used self-collection: other studies have captured this within the Australian program [38,39,40]. Additional research documenting the implementation experience and evolution of these issues overtime, and how they are mitigated, is warranted.

### Implications & Recommendations for Self-Collection Cervical Screening as Health System Tool

Australia is one of a small but growing number of high-income countries to introduce self-collection as a primary screening option (after Argentina and the Netherlands [41,42], with New Zealand also planning to include a universal option to use self-collection from mid-2023 [43]). Around 10 other countries offer self-collection only to under-screened women [41]. The experience in Australia highlights some challenges and disadvantages of this restricted approach. If this approach is used, careful consideration should be given to how overdue people need to be, how eligibility can be ascertained easily, how to ensure under-screened people and healthcare providers are aware that self-collection is an option, and how to deliver the option and test kit. The model used in Australia differs from many other countries, as in Australia providers need to offer people the choice of self-collection vs. clinician-collection; whereas in other settings people who are under-screened are sent information about how to request a self-collection kit, [41] or the test is offered by community health workers [42]. Australia’s policy approach was based on the importance placed on involving a healthcare professional who can facilitate any required follow-up, and results from local pilots showing much higher uptake of self-collection when offered by a provider [39,44] than when offered through a mailout model [43,45]. As more countries consider and plan to move from cytology-based to HPV-based cervical screening, the question of how to best offer self-collection, who to offer it to and how to best prepare the health system for this transition, is becoming increasingly important.

## 5. Conclusions

The availability of self-collection has wide-reaching potential to improve equitable access to cervical screening in many contexts. These findings are particularly valuable for health systems looking to transition to primary HPV testing and to improve program participation. However, as documented in this study, Australia’s experience of introducing self-collection only for under-screened populations highlights the need for robust planning and for consistent, clear communication with stakeholders tasked with implementation. Programs need to ensure that primary care professionals have clarity over eligibility, test accuracy and clinical use, through sufficient resources and education, in order for them to feel confident in offering people the choice of self-collection. Similarly, making sure the pathology sector is equipped and enabled to reliably test self-collected samples is essential to ensuring equitable access within a national screening program. Successful implementation of self-collection in Australia and globally is key to significantly improving cervical screening participation to reach the level required to achieve cervical cancer elimination, and to improve the timeliness and equity of elimination [12,23,46].

## Figures and Tables

**Table 1 ijerph-19-15776-t001:** Definition of implementation outcomes defined in STORIES of self-collection’s initial implementation in the Australian NCSP.

Implementation Outcome	Definition by Proctor [24]	Defined by STORIES (Appendix A)
** Acceptability **	Perception among implementation stakeholders that a given treatment, service, practice, or innovation is agreeable, palatable, or satisfactory	Perception that self-collection cervical screening is satisfactory, palatable, or satisfactory as a health system tool to increase cervical screening within the Australian NCSP
** Adoption **	The intention, initial decision, or action to try or employ an innovation or evidence-based practice	Perception that self-collection would be adopted, employed or utilised by practitioners, pathology providers and eligible people within the Australian NCSP
** Appropriateness **	The perceived fit, relevance, or compatibility of the innovation or evidence-based practice for a given practice setting, provider, or consumer; and/or perceived fit of the innovation to address a particular issue or problem.	The perceived fit, relevance or compatibility of self-collection cervical screening as an evidence-based tool to increase cervical screening amongst under- or never screened populations in the Australian NCSP context
** Feasibility **	Defined as the extent to which a new treatment, or an innovation, can be successfully used or carried out within a given agency or setting	The perception of the extent to which self-collection and the practitioner-supported model of care was successfully implemented within the clinical guidelines in the primary care and pathology sectors
** Fidelity **	Defined as the degree to which an intervention was implemented as it was prescribed in the original protocol or as it was intended by the program developers	The perception of the degree to which self-collection was implemented as intended or prescribed by the Australian NCSP
** Penetration **	Defined as the integration of a practice within a service setting and its subsystems	The perception of the extent to which self-collection cervical screening was integrated into primary care and the wider health system, i.e., tertiary follow-up, colposcopy services
** Implementation Cost **	Defined as the cost impact of an implementation effort	The perception of the implementation cost or incremental cost of implementing self-collection as a health system tool to improve cervical screening within the Australian NCSP
** Sustainability **	Defined as the extent to which a newly implemented treatment is maintained or institutionalized within a service setting’s ongoing, stable operations	The perception that self-collection is maintained and operationalized as routine or an ongoing method within a service setting and broader health system

**Table 2 ijerph-19-15776-t002:** Professional groups of participants’ interviews in the STORIES study.

Professional Group	n
Healthcare provider (e.g., general practitioner (GP), nurse, gynaecologist)	18
Pathology sector	10
Program (Commonwealth, State/Territory, Primary Health Care Network, Registry providers)	11
Policy/Advocacy (e.g., NGOs, professional society)	7
Advisory Committee member (Renewal specific)	8
Education provider	4
Researcher	3
Other **	3

** ‘Other’ category includes: a consumer representative, a medical student and a medical intern.

## Data Availability

The datasets generated and/or analysed during the current study are not publicly available due to the protection of the privacy and confidentiality of interview participants in the STORIES study. Data may be available from the corresponding author on reasonable request if appropriate ethics approvals are put in place. All authors had full access to data for the study.

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
