# Peer review of "“So, if she wasn’t aware of it, then how would everybody else out there be aware of it?”—Key Stakeholder Perspectives on the Initial Implementation of Self-Collection in Australia’s Cervical Screening Program: A Qualitative Study"

_ijerph, 2022, doi:10.3390/ijerph192315776_

Round 1
Reviewer 1 Report
This is topical paper as many countries are considering or even preparing to change to self-collected sampling. The experience in Australia provides real life insight into issues which arose and mitigations which could be made. I would be interested in hearing less about solutions and pre-emptive actions. Conversely, the introduction sets the scene clearly, but I think, in particular round process of self-collection, that this section could be shortened.
Author Response
This is topical paper as many countries are considering or even preparing to change to self-collected sampling. The experience in Australia provides real life insight into issues which arose and mitigations which could be made. I would be interested in hearing less about solutions and pre-emptive actions. Conversely, the introduction sets the scene clearly, but I think,
- The authors thank Reviewer 1 for this comment. We believe most detail included about the process of self-collection is warranted in the introduction because of the unique implementation issues related to self-collection’s process within the Australian National Cervical Screening Program
- Firstly, the model of care described is specific to the Australian National Program context i.e., being overseen by a healthcare professional specially for overdue people within a national screening program; in comparison to other models of care utilised in the literature as described such as mail-out and home-visit models of care for under screened people. Many of the implementation issues described in the results pertain to practitioners; the key implementers of self-collection in the Australian program; being unprepared or confused leading to low-uptake and missed opportunities
- Secondly, the unexpected regulatory requirement to have self-collection testing devices validated prior to its initial roll-out is also a unique implementation issue specific to the Australian context and warrants mentioning as our results relate to this specific issue.
- The restrictive eligibility criteria of self-collection when its initial introduced (available to people over 30 years, who are overdue for screening and refused a clinician-collected test) needs to be explained to contextualize the findings as data was collected in the first 17-months of self-collection’s availability when this eligibility criteria was in place.
Reviewer 2 Report
It was a pleasure reading the paper “So, if she wasn't aware of it, then how would everybody else out there be aware of it?” - Key stakeholder perspectives on the initial implementation of self-collection in Australia’s Cervical Screening Program: A qualitative study”. I believe it addresses an important topic and has the potential to strengthen the area of literature in improving cervical cancer screening in under-screened Australian women by examining the implementation of self-collection methods and how to systematically improve this process. I have included specific recommendations below to strengthen this paper.
Abstract:
Line 9: Define HPV at first mention.
Introduction:
Please review the introduction section for grammar and punctuation errors.
Lines 41-43: This sentence does not make sense as written. Please revise. “This has impart is due a long-term decline in cervical screening participation (5) however, with just over half of the eligible population screening over a 2-year period as recommended (6).”
Lines 52-53: Please add relevant citation for this statement. “Modelling data indicates that Australia is on track to eliminate cervical cancer by 2028-2035…”
Line 72: Did part of the sentence get cut off here? “be highly acceptable to participants and (19, 20).”
Line 87: Should be written as “…due to regulatory requirements…”
Materials & Methods:
It may be worth adding a sentence or two here justifying how a sample size of 49 gives this study adequate information. Did the researchers notice data saturation or were new themes still presented?
I would add a reference to the supplementary document which contains the interview. Maybe add this in parentheses after you mention that it is a semi-structured interview. That way the reader is aware that the interview is available in the supplementary documents.
Lines 129-131: Was the sentence cut off here? Please revise. “No themes related to implementation costs and sustainability were deduced as these outcomes relate to”.
Results:
Please define GP at first mention.
Discussion:
Please review the Discussion for grammar and punctuation.
Line 251: There is a typo here and the question mark should be removed. “the sense of? autonomy”
Line 300: Please revise this sentence. “…implementation in amongst primary care…”
Limitations:
I see a statement about strengths, but please add a statement about the limitations of this study. Being that these results are meaning to speak to the implementation across Australia, it may be relevant to add a statement about participant location…i.e., does the sample cover the target area to achieve the diversity needed for these types of implications?
Conclusions:
No comments
Author Response
It was a pleasure reading the paper “So, if she wasn't aware of it, then how would everybody else out there be aware of it?” - Key stakeholder perspectives on the initial implementation of self-collection in Australia’s Cervical Screening Program: A qualitative study”. I believe it addresses an important topic and has the potential to strengthen the area of literature in improving cervical cancer screening in under-screened Australian women by examining the implementation of self-collection methods and how to systematically improve this process. I have included specific recommendations below to strengthen this paper.
Abstract:
Line 9: Define HPV at first mention.
- Changed to human papillomavirus (HPV)
Introduction:
Please review the introduction section for grammar and punctuation errors.
Lines 41-43: This sentence does not make sense as written. Please revise. “This has impart is due a long-term decline in cervical screening participation (5) however, with just over half of the eligible population screening over a 2-year period as recommended (6).”
- Changed to: This impart is due a long-term decline in cervical screening participation (5) with just over half of the eligible population screening over a 2-year period as recommended (6).
Lines 52-53: Please add relevant citation for this statement. “Modelling data indicates that Australia is on track to eliminate cervical cancer by 2028-2035…”
- The following reference has been added: Hall MT, Simms KT, Lew JB, Smith MA, Brotherton JM, Saville M, et al. The projected timeframe until cervical cancer elimination in Australia: a modelling study. The Lancet Public Health. 2019;4(1):e19-e27
Line 72: Did part of the sentence get cut off here? “be highly acceptable to participants and (19, 20).”
Changed to: Self-collection overcomes many of these barriers, is just as effective in detecting cervical precancer as clinician-collected tests, has shown to increase participation and is highly acceptable to screening participants (19, 20)
Line 87: Should be written as “…due to regulatory requirements…”
- Changed to: Although previous reports have highlighted the delays in the full implementation of self-collection due to regulatory requirements of the testing device as commercially available test was not apparent,….
Materials & Methods:
It may be worth adding a sentence or two here justifying how a sample size of 49 gives this study adequate information. Did the researchers notice data saturation or were new themes still presented?
- The authors thank Reviewer 2 for this comment. The total sampling frame n has been included: n = 87 as well the following sentences justifying the sample size:
- 49 participants represented over half the original sampling frame. The STORIES research team believed 49 interviews generated sufficient data to reflect consistency across the themes deduced and data saturation.
I would add a reference to the supplementary document which contains the interview. Maybe add this in parentheses after you mention that it is a semi-structured interview. That way the reader is aware that the interview is available in the supplementary documents.
- The authors thank Reviewer 2 for this advice. We have referred to supplementary materials; appendix A and B within the methods section to reflect the use of our interview guide and how we operationalized Proctor’s Conceptual Framework for Implementation Outcomes.
Lines 129-131: Was the sentence cut off here? Please revise. “No themes related to implementation costs and sustainability were deduced as these outcomes relate to”.
- The authors thank Reviewer two for this comment. The following sentence has been revised and now reads: No themes related to implementation costs and sustainability were deduced as these out-comes are related to late-stage program implementation as the focus of our research was initial implementation.
Results:
Please define GP at first mention.
- Changed to general practitioner (GP) in Table 2. Professional groups of participants’ interviews in the STORIES study
Discussion:
Please review the Discussion for grammar and punctuation.
Line 251: There is a typo here and the question mark should be removed. “the sense of? autonomy”
- Changed to sense of autonomy
Line 300: Please revise this sentence. “…implementation in amongst primary care…”
- Changed to implementation amongst primary care
Limitations:
I see a statement about strengths, but please add a statement about the limitations of this study. Being that these results are meaning to speak to the implementation across Australia, it may be relevant to add a statement about participant location…i.e., does the sample cover the target area to achieve the diversity needed for these types of implications?
- The authors that Reviewer 2 for this advice. We have amended the following paragraph after highlight strengths to reflect the limitations of the study we have identified. We believe our sampling frame was robust to capture a diversity in the experience of stakeholders involved in the implementation of self-collection as apart of the Australian National cervical screening program. As described in the methods section: “Potential participants were included based on their role in the rNCSP, direct or indirect involvement in the implementation process, their knowledge or expertise and location. Sampling was completed with a view to achieve to broad diversity in participant location and role in the rNCSP”.
- The following sentence has also been added to address further address this aspect of our study: We purposively sampled stakeholders based on their role in implementing self-collection as a part of the rNCSP and location to represent a diverse range of opinions and experiences.
Conclusions:
No comments
- We thank reviewer 3 for their overall feedback on our manuscript
Reviewer 3 Report
The authors have a very important message to get out to the public and while I do not think it is ready for publication, with a little more thought it will be.
I think the overall finding is health system failure: Lack of preparation system wide to ensure all the parts of the system are in alignment and ready for implementation. For example, poor communication within and across the health system (rNCSP) resulting in laboratories not being prepared, health services not being prepared etc. Health system failure impacted on the health and well being of minority group’s in particular indigenous and underprivileged cultures, a very serious offense (is this really about institutional racism?).
The framework used in the methodology was useful. Presenting the findings following the same sequence would make the findings easier to follow. The discussion could draw on specific system level or lack of system level direction – policies etc. and tie this into the findings.
It would be worth while getting a proof reader in. There are several mistakes in the text – ie lines, 11, 22, 41, 131, 269. Some formatting mistakes in the reference section – needs checking.
Author Response
The authors have a very important message to get out to the public and while I do not think it is ready for publication, with a little more thought it will be.
I think the overall finding is : Lack of preparation system wide to ensure all the parts of the system are in alignment and ready for implementation. For example, poor communication within and across the health system (rNCSP) resulting in laboratories not being prepared, health services not being prepared etc. Health system failure impacted on the health and well being of minority group’s in particular indigenous and underprivileged cultures, a very serious offense (is this really about institutional racism)
- The authors thanks Reviewer 3 for their insightful comment. The authors acknowledge that whilst institutional racism is a prominent and pervasive issue within the Australian health system; we believe that is not the causal issue of why self-collection was not initially implemented as intended. Our results reflect lack of preparation and transparency regarding the introduction of self-collection. We agree that the initial implementation of self-collection was a health system failure which had follow-on effects to how service providers utilised and trusted self-collection as new innovation; however, this manuscript does not focus on how this failure impacted services provision to priority populations such as Aboriginal and Torres Strait Islander women or cultural and linguistically diverse populations; this was not reflected specifically in this sub-analysis of the STORIES study and is thus reflected in another sub-analysis with this specific focus.
- We agree however that the health system needed to do better in involving important community stakeholders in the initial implementation planning to ensure the needs of these priority populations were met, which would in turn maximize uptake of self-collection. This is in itself, a health system failure as well. This consideration is reflected within an overarching manuscript of the STORIES study as it pertains to multiple program changes brought about the transition of the rNCSP i.e., the transition from cytology-based to HPV-DNA based cervical screening.
The framework used in the methodology was useful. Presenting the findings following the same sequence would make the findings easier to follow. The discussion could draw on specific system level or lack of system level direction – policies etc. and tie this into the findings
- The authors thank Reviewer 3 for this praise and advice. The results section has outlined the results in which domains according to Proctor were coded for the most. This is currently outlined in theme headings i.e. theme 1) acceptability and appropriateness of self-collection; theme 2) feasibility and fidelity of self-collection, theme 3) the impact of promotion on adoption and penetration of self-collection. We acknowledge that relevant domains were grouped together to allow ease of writing. We have also removed the table from the last domain and written this within the body of the manuscript for clarity and flow.
- The authors have addressed the lack of health system direction and have addressed this throughout the discussions and linked to our findings.
It would be worth while getting a proof reader in. There are several mistakes in the text – ie lines, 11, 22, 41, 131, 269. Some formatting mistakes in the reference section – needs checking.
- We thank Reviewer 3 for this comment. The manuscript has been proofread by the corresponding author with the afore mentioned errors corrected in addition to checking the reference section.
Round 2
Reviewer 3 Report
Improvement on the original submission. I think this paper will be helpful to other countries trying to implement a self-sampling HPV programme.